# Effective BiOCl Electrons Collector for Enhancing Photocarrier Separation of Bi$_2$WO$_6$/BiOCl Composite

**Yi Zheng** [1], **Siqi Wang** [1], **Min Shu** [1], **Yi Wang** [1] **and Dumeng Cao** [2,3,*]

1 School of Petrochemical Technology, The Lanzhou University of Technology, A Pengjiaping Road No. 36, Lanzhou 730050, China; wind_2000_love@163.com (Y.Z.); wsq7770426@163.com (S.W.); shumin_1122@163.com (M.S.); wangyi@lut.edu.cn (Y.W.)
2 National Nickel and Cobalt Advanced Materials Engineering Research Center, Lanzhou 730050, China
3 Lanzhou Jinchuan Technology Park Co., Ltd., Lanzhou 730050, China
* Correspondence: csucdm@163.com

**Abstract:** Enhancing photocarrier separation is a key step of photocatalysis, and in situ constructed composition interface is an advanced method to achieve this aim. Therefore, we report a face-to-face Bi$_2$WO$_6$/BiOCl (BWOC) which was synthesized via the continuous in situ ion-exchange method. As UV light is harmful to the human body, BWOC exhibits excellent photocatalytic activity only in visible light, and this is an important feature because visible light is a human-friendly operating condition. Under 50 W visible LED lamp illumination, unexcited BiOCl (BOC) only extracts electrons of excited Bi$_2$WO$_6$ (BWO), and holes remain on BWO, resulting in excellent photocarrier spatial separation efficiency through the face-to-face interface. This is why BWOC can be safe to use for the removal of hazardous substances. Compared with BWO and BOC, BWOC possesses 2.6 and 5.6 times higher photodegradation activity than RhB. This work provides a novel insight of efficient visible light photocatalytic system for environmental remediation.

**Keywords:** Bi$_2$WO$_6$; BiOCl; heterojunction; photocatalysis





## 1. Introduction

With the deepening of social development, protection of the daily contaminated living space becomes a research hotspot. Using the chemical method to address issues has more benefits than physical ways. Therefore, many advanced functional materials have been synthesized via chemical reaction. Photocatalysis is considered the most promising chemical method for environmental remediation and energy regeneration through the redox reaction [1–3]. Many photocatalysts have been studied for this aim, e.g., TiO$_2$, [4,5], CdS [6,7], MoS$_2$ [8,9], CeO$_2$ [10,11], etc. However, all kinds of intrinsic photocatalysts have certain problems that need to be resolved, such as poor light absorption, low separation of photocarriers, etc. Therefore, the development of modification research to meet the requirements of potential industrial applications is an essential part of a scientist's work life.

Bismuth-based photocatalysts such as Bi$_2$WO$_6$ [12], BiOCl [13], BiVO$_4$ [14], Bi$_2$S$_3$ [15], etc., have been used as model materials due to the same unique layered crystal unit, which can be used for construction chemical-bonded interface. Bi$_2$WO$_6$, as one of the representative members of bismuth-based photocatalysts, is a famous and efficient photocatalyst for the degradation of organic pollutants. Owing to its suitable oxidation potential and good visible light response, it has attracted wide attention. Although the internal electric field in a crystal is advantageous for the transport of bulk photocarriers, the spatial separation of carriers is still required to prevent carrier recombination [16]. Coupling them with other bismuth-based photocatalysts is a good strategy to improve the carrier separation efficiency of Bi$_2$WO$_6$. To construct Bi$_2$WO$_6$-based composites with high-quality contact for enhancing photocarrier separation of Bi$_2$WO$_6$, ion-exchange is used, and BiOCl is used

as the matrix, due to the same crystal unit, suitable band structure, and matching crystal lattice [17]. However, there are few reports about $Bi_2WO_6$ compounding with BiOCl via the ion-exchange method. Therefore, more investigations are needed.

Herein, a face-to-face 2D $Bi_2WO_6$/2D BiOCl composite was synthesized. The synthesized samples were characterized by SEM, TEM, XRD, and XPS, confirming the formation of the heterojunction. Compared with $Bi_2WO_6$ and BiOCl, the composite photocatalyst exhibited a prominently photocatalytic performance in the degradation process of RhB. PL, photoelectrochemical tests, and band structure show that BiOCl is an electron collector to enhance photo-holes' concentration on $Bi_2WO_6$, leading to an increase in the degradation time to around 3 times higher compared to $Bi_2WO_6$.

## 2. Experimental Section

### 2.1. Materials

$Bi(NO_3)_3 \cdot 5H_2O$ was AR purity and bought from Aladdin Inc. (Shanghai, China). The other reagents were also AR grade, and all were purchased from Sinopharm Chemical Reagent Co., Ltd. (Tianjin, China). All reagents were directly used without further purification.

### 2.2. Synthesis of Photocatalysts

$Bi_2WO_6$/BiOCl was prepared by the one-step hydrothermal method. The detailed steps are as follows: 1 mmol $Bi(NO_3)_3 \cdot 5H_2O$ was dissolved in 33 mL ultrapure water, stirred, and dispersed sufficiently; then $x$ amount of $NH_4Cl$ ($x$ = 0, 0.11, 0.25, 0.43) was added under continuous stirring; finally, 0.5 mmol $Na_2WO_4$ $2H_2O$ was added and stirred for 30 min. The white suspension was transferred to a 50 mL Teflon-lined autoclave, sealed, and heated for 24 h at 160 °C. The product was collected after the autoclave was cooled to room temperature, washed several times with ultra-pure water and ethanol, and dried for 10 h at 60 °C. The products synthesized under different amounts of $NH_4Cl$ (0, 0.11, 0.25, and 0.43 mmol) were labeled BWO, BWOC1, BWOC2, and BWOC3, respectively. In addition, pure BiOCl was also synthesized by the same hydrothermal method without adding $Na_2WO_4 \cdot 2H_2O$.

### 2.3. Characterization of Samples

The crystal structure of the samples was characterized by a powder X-ray diffractometer (XRD, RigakuD/max-2000), which is equipped with a Cu-Ka radiation at a $10° \text{ min}^{-1}$ scanning rate and in the angle range of 10–80°. The morphology and crystal information were observed by scanning electron microscopy (SEM, JEOL (Akishima-shi, Japan) JSM-6700F) and transmission electron microscopy (TEM, JEOL (Akishima-shi, Japan) JEM-2100F). X-ray photoelectron spectroscopy (XPS) was measured for a chemical environment of elements by a Thermo Scientific ESCA Lab250 spectrometer (Thermo Scientific, Waltham, MA, USA), which consists of a monochromatic Al Kα as the X-ray source (obtained data were calibrated by the C1 speak at 284.8 eV). UV–Vis diffuse reflectance spectroscopy (UV–Vis DRS) was studied on an ultraviolet–visible spectrophotometer (UNICO (Shanghai, China) 2800-A), and $BaSO_4$ was taken as reflectance standard material. AUTO LAB electrochemical workstation (PGSTAT302N) with three-electrode was employed to bring the photoelectrochemical responses to room temperature; the Pt wire was a counter electrode, Ag/AgCl was the reference electrode, 0.25 M $Na_2SO_4$ was an electrolyte, and the light source was 300 W Xe lamp (Perfect Light PLS-SXE300) with UV cutoff.

### 2.4. Photocatalytic Activity Test

The photocatalytic activity of the prepared photocatalysts was evaluated by degradation of RhB under a 50 W LED lamp ($\lambda \geq 420$ nm). Then 50 mg of photocatalyst was dispersed in 100 mL RhB solution with 5 mg/L, and the mixture was put in ultrasonic for 3 min and stirred in the dark for 30 min. After adsorption and turning on a light, a 2 mL solution was taken at regular intervals and centrifuged, and the absorbance of the supernatant was analyzed by a UV–Vis spectrophotometer.

## 3. Results and Discussion

The XRD patterns of the prepared samples can be seen in Figure 1. The diffraction peaks of BWO and BOC could be well indexed to the orthorhombic BWO (JCPDS: 73–2020) and tetragonal BOC (JCPDS: 85–0861). BWOC composites exhibit the coexistence of both orthorhombic BWO and tetragonal BOC. No other peaks are observed, thus indicating that there was no impurity. Here, the mole ratio of Bi and Cl is 2.34~9.07 (Cl$^-$ excess), while the BWO is generated after adding $Na_2WO_4$ (mole ratio of Bi and W = 2), thus indicating that the solubility of BWO is less than BOC and that precipitation conversion has occurred. In addition, the content of BOC gradually increased with the increase of $NH_4Cl$, suggesting that the precipitation conversion is inhibited by high chloride concentration. When the precipitation equilibrium is reached, BWOC composite is obtained. Inversely, if BWO is synthesized first, BOC cannot be found in the XRD pattern when adding Cl$^-$ (as shown in Supplementary Figure S1), thus demonstrating that crystalline BWO does not readily convert to BOC.

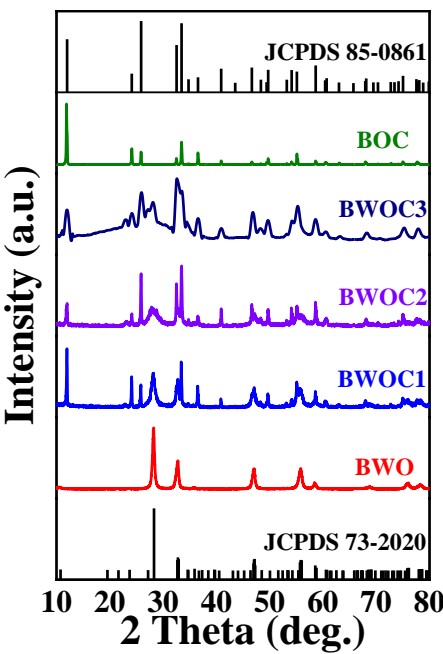

**Figure 1.** XRD patterns of the prepared samples: $Bi_2WO_6$(BWO), $Bi_2WO_6$/BiOCl composites (BWOC1–3), and BiOCl (BOC).

The morphology of the samples was characterized by SEM. Figure 2a,c shows the morphologies of $Bi_2WO_6$ and BiOCl. It can be seen that $Bi_2WO_6$ is an even sphere-like hierarchical structure which is composed of many stacked nanosheets. BiOCl is a lamellar structure with an average diameter of about 2.0 μm. The morphology of the $Bi_2WO_6$/BiOCl composite retains that of $Bi_2WO_6$ with a certain reduction in size, as shown in Figure 2b. This may be due to precipitation conversion; some BiOCl may act as $Bi^{3+}$ donors for the formation of $Bi_2WO_6$; thus, the low concentration of $Bi^{3+}$ leads to a small size of $Bi_2WO_6$, and the consumption of BiOCl also reduces the size of BiOCl. In addition, such precipitation conversion causes $Bi_2WO_6$ to grow in situ on BiOCl, with a face-to-face relationship, which can be seen in Figure 2d and Supplementary Figure S2. Generally, in situ growth can build a high-quality interface for the transmission and separation of photocarriers [18]. The lattice spacing of BiOCl is about 0.368 nm, which is consistent with the (002) plane. The marked lattice spacings of 0.375 nm appearing on the larger sheet-shaped sample correspond to (111) planes of $Bi_2WO_6$.

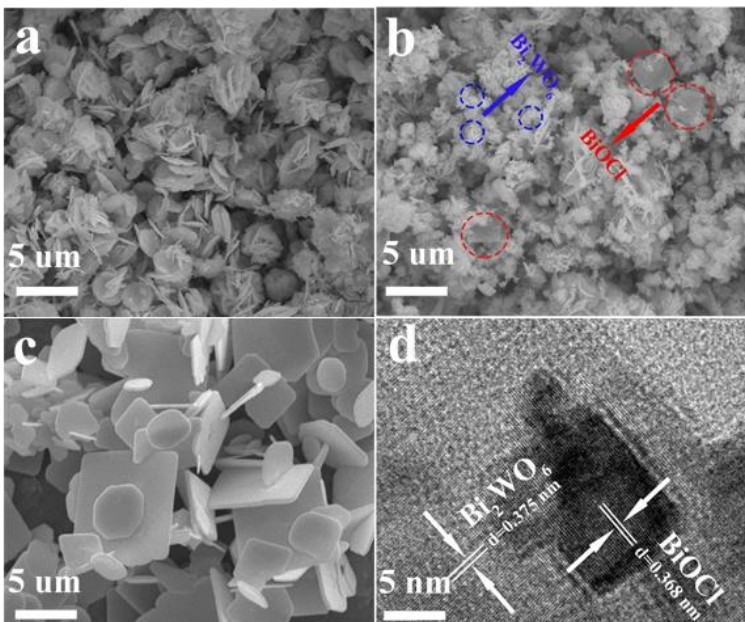

**Figure 2.** (**a**) SEM and TEM images of as-prepared samples: SEM image of $Bi_2WO_6$, (**b**) BWOC2($Bi_2WO_6$/BiOCl composite samples), and (**c**) BiOCl; (**d**) HRTEM image of BWOC2. Bi2WO6 is marked in blue color, BiOCl is marked in red color.

The light-response capacity of the as-prepared photocatalysts was investigated by using UV–Vis DRS, as shown in Figure 3a. The absorption edges of mono BWO and BOC are 443 and 379 nm, respectively. For the composites, the absorption edges of composites are same, showing that the light response range is same. In addition, BWOCs can absorb more visible light from 475 to 675 nm than BWO (BWO cannot absorb visible light in this region), and this may be due to the chemical interaction at the interface. Among BWOCs, BWOC2 displays better photo-response in the range of 475 to 675 nm, due to the suitable composite proportion, thus suggesting that BWOC2 may form more photocarriers than others.

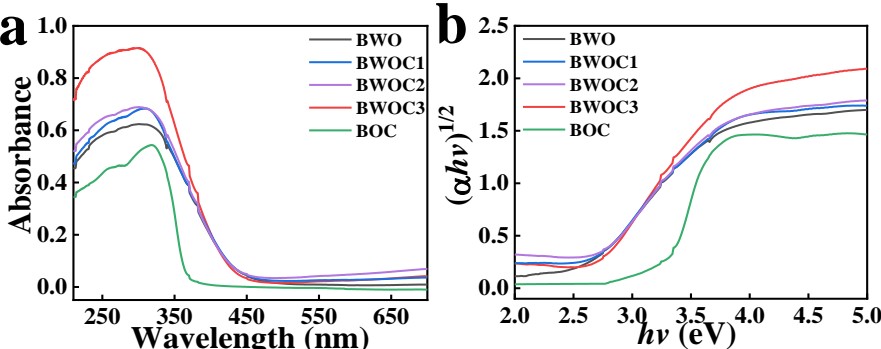

**Figure 3.** (**a**) UV–Vis diffuse reflectance spectra and (**b**) Tauc curves of the as-prepared samples.

The bandgap energy of photocatalysts was obtained from the following formula:

$$(\alpha h v)^{1/n} = A(h v - E_g)$$

where $\alpha$ is the absorption coefficient, $v$ is the frequency of vibration, $h$ is Planck's constant, $E_g$ is the bandgap energy, and n is 1/2 for BWO (direct transition) or 2 for BOC (indirect transition) [19,20]. According to this formula, the bandgap of BWO is calculated to be about 2.70 eV, indicating that BWO can respond to visible light. The bandgap of BOC is 3.29 eV, which can only be excited in the ultraviolet region.

BWOC2 was further studied by XPS. The binding energy in the spectrum was calibrated by using C 1s (284.62 eV) [21]; all peaks on the XPS curve for the composite were attributed to Bi 4f, W 4f, Cl 2p, and O 1s (Figure 4a). In Figure 4b, there are two prominent peaks at 159.0 and 164.3 eV contributing to Bi $4f_{7/2}$ and Bi $4f_{5/2}$, thus confirming that the bismuth species in the BWOC2 is $Bi^{3+}$ [22]. The W 4f high-resolution XPS spectrum shows two principal peaks with binding energy at about 34.9 and 37.0 eV, corresponding to $W^{6+}$ in the samples (Figure 4c) [23]. In addition, the peaks centering at 531.0 and 531.9 eV can be designated to the coordination of oxygen in Bi–O and oxygen in W–O (Figure 4d) [24].

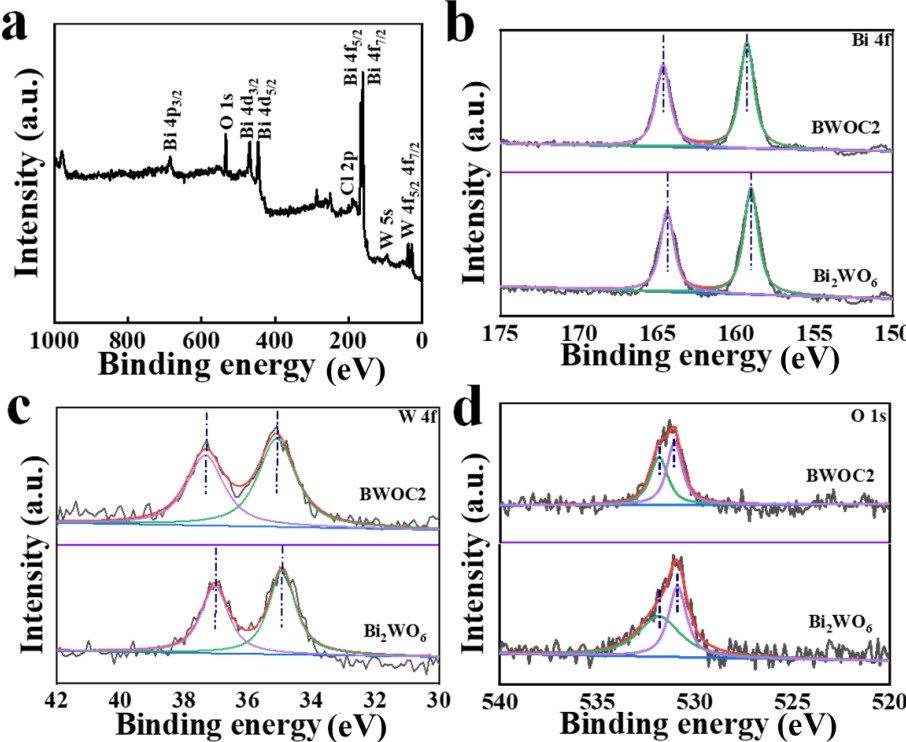

**Figure 4.** XPS spectra of BWOC2 sample: (**a**) survey spectrum, (**b**) Bi 4f, (**c**) W 4f, and (**d**) O 1s.

If the peaks of Bi $4f_{7/2}$ and Bi $4f_{5/2}$ in BWOC2 shift to higher binding energy compared with BWO, the same goes for W peaks. The result is that the electronic density of $W^{6+}$ and $Bi^{3+}$ decreases, and this induces the increase of binding energy of W 4f and Bi 4f.According to the results of HRTEM of BOC and BWO, oxygen and chlorine atoms are at the end of the (002) plane of BOC and tungsten and bismuth atoms are at the end of the (111) facet of BWO. When the (002) plane of BOC is perpendicular to the (111) facet of BWO, the oxygen atom of BOC may be close to the tungsten atom of BWO, and the chlorine atom of BOC may be near to the bismuth atom of BWO.

The photocatalytic performance of the as-prepared samples was evaluated by the degradation of RhB solution under visible light irradiation ($\lambda \geq 420$ nm). Photodegradation curves of as-prepared samples are shown in Figure 5a. After 60 min, 70% RhB was eliminated by the self-sensitized process, [25] and 90% RhB was degraded by BWO. Compared to BOC and BWO, 100% RhB can be removed by BWOC2 just by using 20 min. The apparent rate constant, $k$, of BWOC2 is 0.09605 $min^{-1}$, which reaches 2.62 and 5.58 times as much as those of BWO (0.03664 $min^{-1}$) and BOC (0.01721 $min^{-1}$), as shown in Figure 5b [26,27]. Furthermore, the photostability of BWOC2 is researched by recycling experiments, and the photocatalytic activity remains well after four recycles (Figure 5c), suggesting the high stability of the sample.

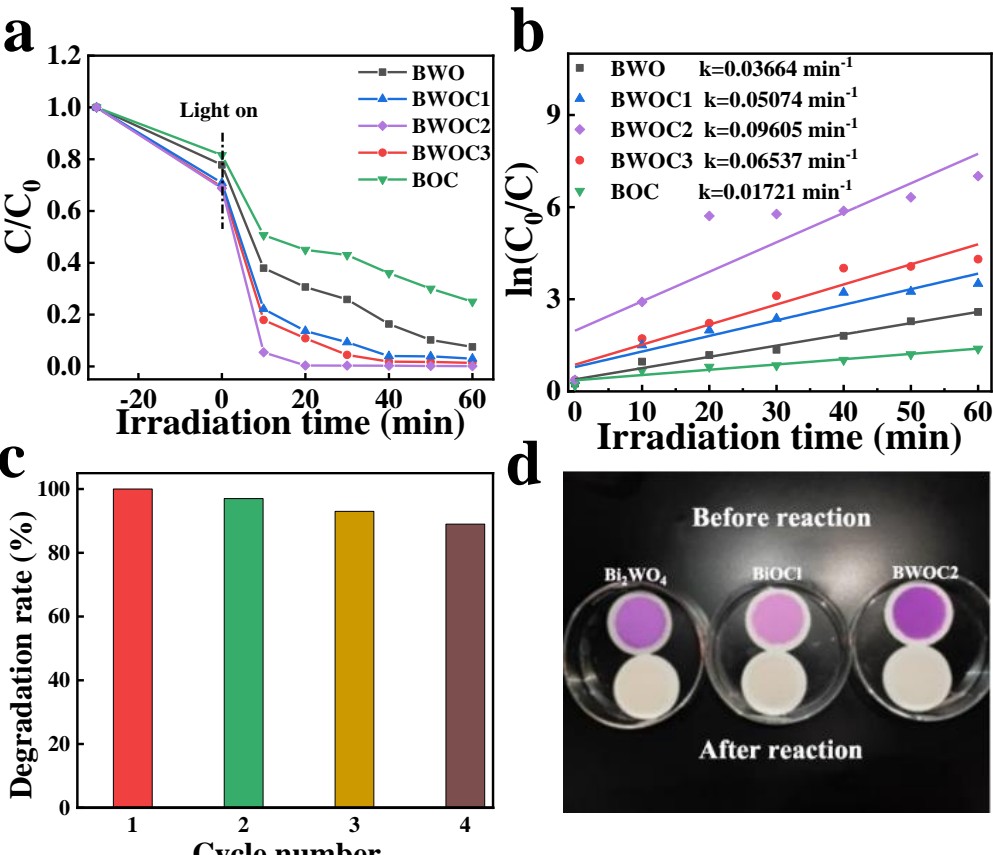

**Figure 5.** (**a**) Photodegradation curves, (**b**) $\ln(C_0/C)$ vs. time curves and rate constant, *k*, over different samples, (**c**) recycling tests of BWOC2 sample, and (**d**) color contrast of BWOC2 sample before and after the reaction.

Meanwhile, we compared the XRD patterns of BWOC2 samples before and after the four cycles. It was found that the position, intensity, and half-peak width of characteristic peaks of the BWOC2 samples did not change significantly before and after the cycles, thus further proving that the BWOC2 samples had good stability (Supplementary Figure S3). The dark adsorption of BWOC2 is larger than that of BWO and BOC. The larger adsorption is a benefit to the photodegradation, but if RhB remains on the surface of the catalyst, it cannot deem that the catalyst can degrade the pollutant completely. Therefore, the color of the catalyst before and after the reaction is recorded (Figure 5d), and the color of the photocatalysts changes to the original color after the reaction ended, thus confirming that the as-prepared sample can thoroughly remove RhB from both the solution and catalyst surface.

To understand the reasons for the improvement of photocatalytic performance, PL was firstly studied, as shown in Figure 6a. It displays that the PL intensity of BWOC is lower than that of pure BWO and BOC. A lower PL intensity indicates that the recombination of photogenerated electron–hole pairs of photocatalysts is suppressed [28,29]. Under visible light illumination, BWOC2 owns a higher photocurrent density, indicating an efficient separation of photocarriers (Figure 6b) [30,31]. The photocurrent density of BWOC2 decreased and remained stable after 400 s, indicating that BWOC2 can generate a large transient current and reach a steady state after 400 s. The photocurrent density of the samples was reproducible during the repeated on/off cycle for certain times, thus indicating that the prepared catalysts are stable [32]. EIS was further investigated, as shown in Figure 6c. The arc radius of the BWOC2 is the smallest, demonstrating that BWOC2 has the lowest interface resistance, and this may be due to its having a more effective concentration of photocarriers [33,34]. The equivalent circuit diagram of sample is show in Supplementary Figure S4. Rs rep-

resents solution resistance, Rct represents the charge transfer resistance, and CPE is the constant phase angle element. The BWOC2 displayed lower charge transfer resistance compared to the BWO and BOC. The specific information is shown in Supplementary Table S1. Because the position of the conduction band (CB) is close to the flat band potential, the Mott–Schottky test (M–S) can be used to obtain the CB position (Supplementary Figure S2). The CB position of BWO and BOC is $-0.44$ and $-0.4$ V. Combined with the $E_g$ of BWO and BOC, the valence band (VB) position is 2.26 and 2.89 V, respectively.

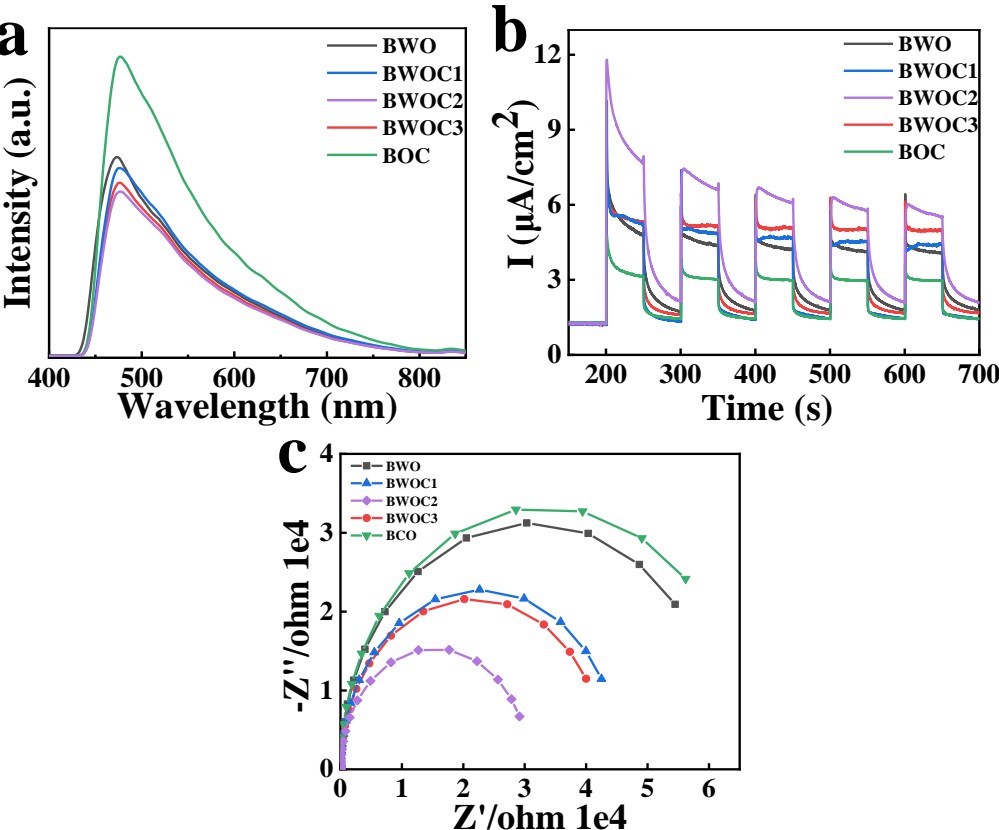

**Figure 6.** (**a**) PL, (**b**) I-t curve, and (**c**) electrochemical impedance spectroscopy of the as-prepared samples.

Based on the above results, it can be understood that BWO can excited by visible light irradiation and forms excitons. Then the photoelectrons will be collected by BOC under the built-in electric field, leaving photo-holes in BWO. As a result, a higher photo-hole concentration on BWO will enhance the oxidation process of organic pollutants.

## 4. A Possible Mechanism for Photocatalytic Degradation of Rhodamine

To understand the photocatalytic mechanism, the trapping agents are added to quench the active species during the photodegradation process. Injections of nitrogen, sodium oxalate (SO), and isopropyl alcohol (IPA) are used as efficient scavengers for $\cdot O_2^-$, $h^+$, and $\cdot OH$ [35,36]. As shown in Figure 7, the photodegradation rate was almost not changed when $N_2$ was pumped, while the photocatalytic activity of BWOC2 was significantly diminished by SO and IPA, suggesting that $\cdot O_2^-$ acts inappreciable role in the photocatalytic process, and yet the main active species are $h^+$ and $\cdot OH$. In addition, the photogenerated hole plays the main role in an anaerobic environment, even if the consumption of excited electrons with $O_2$ is restricted, indicating that the electrons on BWO will transfer to BOC (act as electron collector), and the holes on BWO can still work.

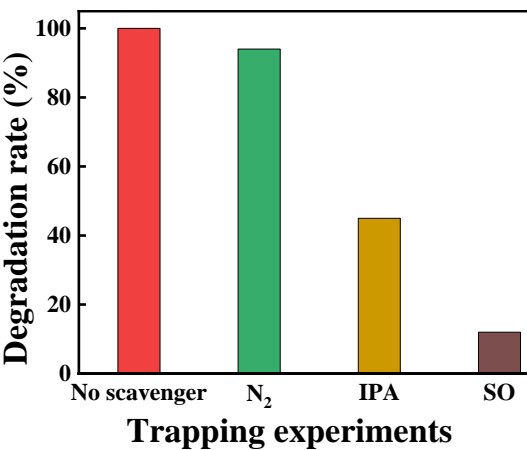

**Figure 7.** Trapping experiments of BOWC2.

According to the above results, a possible photocatalytic mechanism is proposed; the schematic diagram is shown in Figure 8. Due to the bandgap of BOC being 3.29 eV, it cannot be only excited by an ultraviolet LED lamp, but the $Bi_2WO_6$ ($E_g$ = 2.70 eV) can respond to visible light. When BWOC2 is irradiated by visible light, a photogenerated transfer takes place from BWO to BOC, causing the efficient separation of electron–hole pair and highly improving the photocatalytic activity. The CB of BOC is lower than −0.33 eV [37], which cannot be combined with $O_2$ to form $\cdot O_2{}^-$; thus, $\cdot O_2{}^-$ has a minimal effect. The VB level of BWO is lower than 2.18 eV [37], suggesting that the hole on BWO can react with OH⁻ or $H_2O$ to generate the $\cdot OH$ and can oxidation pollutants directly due to the strong oxidation ability [38]. This agrees with the photocatalytic mechanism experiment: the hole and $\cdot OH$ act as the main acted spices, and the hole is more useful than $\cdot OH$.

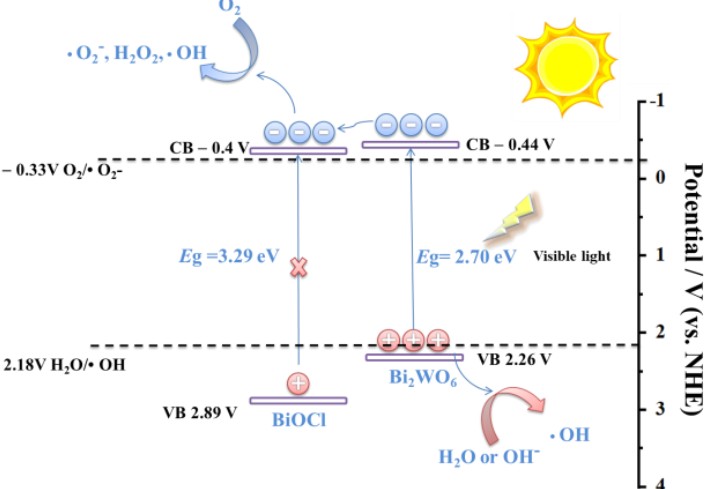

**Figure 8.** Possible photocatalytic mechanism of BWOC2.

## 5. Conclusions

In summary, a face-to-face-type BWOC composite system was successfully constructed; the photocatalyst exhibits much enhanced photocatalytic activity in the degradation of RhB under visible light irradiation. This enhancement is attributed to the separation of the electron on BWO to BOC, thus improving the concentration of holes on BWO for oxidation of RhBs. This work provides some theoretical foundation to the design of a photocatalyst with the non-full visible light response and that can be used for water purification application and environmental remediation.

**Supplementary Materials:** The following supporting information can be downloaded at: https://www.mdpi.com/article/10.3390/chemistry4030054/s1. Figure S1: XRD patterns of the prepared samples: Bi2WO6 (BWO), Bi2WO6/BiOCl composites (BWOC1-3) and BiOCl (BOC). Figure S2: SADE of BWOC2 sample. Figure S3: MS of Bi2WO6 sample and BiOCl sample. Figure S4: XRD patterns before and after cyclic experiment. Figure S5: The equivalent circuit diagram of sample. Table S1: EIS fitting data of sample. Table S2: The corresponding report of Bi2WO6, BiOCl and composite photocatalysts for RhB degradation [39–44].

**Author Contributions:** The investigation, S.W. and M.S.; writing—original draft preparation, Y.Z.; writing—review and editing, Y.W.; visualization, D.C. All authors have read and agreed to the published version of the manuscript.

**Funding:** This work was financially supported by projects of the Lanzhou University of Technology Doctoral Fund (061701) and The National Youth Science Fund Project of the National Natural Science Foundation of China (21802060).

**Institutional Review Board Statement:** Not applicable.

**Informed Consent Statement:** Not applicable.

**Data Availability Statement:** Not applicable.

**Conflicts of Interest:** The authors declare no conflict of interest.

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
