# Peer review of "Effective BiOCl Electrons Collector for Enhancing Photocarrier Separation of Bi2WO6/BiOCl Composite"

_chemistry, doi:10.3390/chemistry4030054_

Round 1
Reviewer 1 Report
The submitted work reports synthesis of BWOC composites. The composite photocatalysts exhibited much enhanced photocatalytic activity in the degradation of RhB under visible light irradiation; this enhancement is attributed to the separation of electron on BWO to BOC, thus improving the concentration of holes on BWO for oxidation of RhB. The topic is of potential interest. Several technical issues are shown herein before it can be accepted for publication.
1. The detailed crystallographic features of the composites should be identified from selected area electron diffraction (SAED) patterns obtained in the TEM measurements.
2. Could authors inform the difference in light harvesting ability among various BWOC composites from their UV-vis analysis results.
3. From the I-t plots, does BWOC2 have photocorrosion effect?
4. The interfacial transfer resistance (Rct) in the composites should be evaluated from the Nyquist plots by constructing a possible equivalent circuit.
5. A table includes a comparison of photodegradation ability of similar composite system from literatures should be provided for clarity.
6. More recycling test numbers should be provided herein to check the stability of the as-synthesized photocatalysts. The detailed microstructures of the photocatalysts after several test cycles should be provided to support the stability of the as-synthesized samples.
7. The proposed photocatalytic mechanism herein is not convincing because CB position is not precisely evaluated from their synthesized reference samples with the Mott-Schlocky (M-S) measurements. Please provide real CB and VB positions of BIOCl and Bi2WO6 from the measurements of M-S and UV-vis, and revised the possible photocatalytic mechanism.
Author Response
Dear Editor,
Thank you very much for your email message dated 17-June-2022 concerning our manuscript entitled “Effective BiOCl electrons collector for enhancing photocarrier separation of Bi2WO6/BiOCl composite” (chemistry-1769902). We also appreciate the comments from the referees.
We have revised the manuscript carefully on the basis of referees' comments and the corresponding changes in the revised manuscript are marked in yellow.
We have also prepared responses to the comments made by the referees, and the detail is listed in the following pages. We hope that the revised manuscript is acceptable for publication in Chemistry.

Reviewer 2 Report
The manuscript titled Effective BiOCl electrons collector for enhancing photocarrier separation of Bi2WO6/BiOCl composite reports a face-to-face relationship providing an efficient carrier separation channel which acts as a photogenerated electrons collector of BWO compared to the dye sensitization center and leads to a higher effective photogenerated holes concentration of BWO aiming to enhance the photoactivity.
The paper has a relevant and original aspect, which is a little bit hard to distinguish in the abstract.
Please, kindly re-write the abstract so that the innovative component of the study is emphasized.
Some suggestions and corrections in your work will make your manuscript more attractive and suitable to be published in Chemistry.
Firstly, some polishing is required in order to merit the publication level of this journal. Authors should revise the paper under the following amendments.
Please, pay attention in terms of English Grammar:
Please identify a clearer and more coherent objective of the research. Please highlight the key points, as well.
1. What is the importance and scope of this work in the context of this journal? Need to discuss the relation/impact of the proposed work with the scope of Chemistry in introduction briefly.
2. Abstract would be kept simple and precise with the novelty of the paper.
Author Response
Dear Editor,
Thank you very much for your email message dated 17-June-2022 concerning our manuscript entitled “Effective BiOCl electrons collector for enhancing photocarrier separation of Bi2WO6/BiOCl composite” (chemistry-1769902). We also appreciate the comments from the referees.
We have revised the manuscript carefully on the basis of referees' comments and the corresponding changes in the revised manuscript are marked in yellow.
We have also prepared responses to the comments made by the referees, and the detail is listed in the following pages. We hope that the revised manuscript is acceptable for publication in Chemistry.
Responses (R) to Referees’ Comments (C)
First of all, we would like to thank the referees for their valuable comments on our manuscript and the following are the responses to the referees’ comments.
Referee: 2
Comment (1): What is the importance and scope of this work in the context of this journal? Need to discuss the relation/impact of the proposed work with the scope of Chemistry in introduction briefly.
Answer:
We rewrote the article and made it more streamlined, with the changes highlighted in yellow.
Comment (2): Abstract would be kept simple and precise with the novelty of the paper.
Answer:
We rewrite the summary.
Round 2
Reviewer 1 Report
The manuscript has been revised according to review comments. It is acceptable for publication.